# Population-Level SARS-CoV-2 RT–PCR Cycle Threshold Values and Their Relationships with COVID-19 Transmission and Outcome Metrics: A Time Series Analysis Across Pandemic Years

**DOI:** 10.3390/v17010103

**Published:** 2025-01-14

**Authors:** Judith Carolina De Arcos-Jiménez, Ernestina Quintero-Salgado, Pedro Martínez-Ayala, Gustavo Rosales-Chávez, Roberto Miguel Damian-Negrete, Oscar Francisco Fernández-Diaz, Mariana del Rocio Ruiz-Briseño, Rosendo López-Romo, Patricia Noemi Vargas-Becerra, Ruth Rodríguez-Montaño, Ana María López-Yáñez, Jaime Briseno-Ramirez

**Affiliations:** 1State Public Health Laboratory, Zapopan 45170, Mexico; judith.dearcos@academicos.udg.mx (J.C.D.A.-J.); titinaquintero2@gmail.com (E.Q.-S.); rlopezromo08@gmail.com (R.L.-R.); 2Laboratory of Microbiological, Molecular, and Biochemical Diagnostics (LaDiMMB), Tlajomulco University Center, University of Guadalajara, Tlajomulco de Zuñiga 45641, Mexico; patricia.vargas@cutlajomulco.udg.mx; 3Antiguo Hospital Civil de Guadalajara, “Fray Antonio Alcalde”, Guadalajara 44280, Mexico; pedro.martinez@cucs.udg.mx (P.M.-A.); roberto.damian@cutlajomulco.udg.mx (R.M.D.-N.); 4Nuevo Hospital Civil de Guadalajara, “Dr. Juan I. Menchaca”, Guadalajara 44340, Mexico; grosales@hcg.gob.mx; 5Health Division, Tlajomulco University Center, University of Guadalajara, Tlajomulco de Zuñiga 45641, Mexico; oscar.fernandezdiaz@academicos.udg.mx (O.F.F.-D.); mariana.ruiz@academicos.udg.mx (M.d.R.R.-B.); ruth.rodriguez@academicos.udg.mx (R.R.-M.); ana.lopez@academicos.udg.mx (A.M.L.-Y.)

**Keywords:** SARS-CoV-2 cycle threshold (Ct), COVID-19 transmission dynamics, time series analysis, respiratory viral trends, epidemiological predictors, cycle threshold (Ct) predictive value

## Abstract

This study investigates the relationship between SARS-CoV-2 RT–PCR cycle threshold (Ct) values and key COVID-19 transmission and outcome metrics across five years of the pandemic in Jalisco, Mexico. Utilizing a comprehensive time-series analysis, we evaluated weekly median Ct values as proxies for viral load and their temporal associations with positivity rates, reproduction numbers (Rt), hospitalizations, and mortality. Cross-correlation and lagged regression analyses revealed significant lead–lag relationships, with declining Ct values consistently preceding surges in positivity rates and hospitalizations, particularly during the early phases of the pandemic. Granger causality tests and vector autoregressive modeling confirmed the predictive utility of Ct values, highlighting their potential as early warning indicators. The study further observed a weakening association in later pandemic stages, likely influenced by the emergence of new variants, hybrid immunity, changes in human behavior, and diagnostic shifts. These findings underscore the value of Ct values as scalable tools for public health surveillance and highlight the importance of contextualizing their analysis within specific epidemiological and temporal frameworks. Integrating Ct monitoring into surveillance systems could enhance pandemic preparedness, improve outbreak forecasting, and strengthen epidemiological modeling.

## 1. Introduction

The global coronavirus disease 2019 (COVID-19) pandemic has highlighted the critical importance of diagnostic tools, particularly real-time reverse-transcription polymerase chain reaction (RT–PCR) tests, in both individual patient management and epidemiological surveillance [1,2,3,4,5]. Cycle threshold (Ct) values, which are inversely correlated with viral load, have been linked to a higher likelihood of culturable virus and increased infectivity and disease severity [5]. These associations underscore their utility in assessing transmission risks and clinical outcomes [6,7,8,9,10]. Moreover, the predictive value of population-level Ct values has emerged as a vital area of study within the COVID-19 pandemic context [11]. As semi-quantitative markers of viral load, Ct values not only offer insights into individual infectiousness but also serve as valuable tools at a population level for monitoring viral transmission, predicting outbreaks, and guiding public health interventions [12,13].

From an epidemiological perspective, Ct values function as proxies for viral transmissibility, with declines across a population often signaling increased community-level transmission [12,13]. These shifts can precede observable increases in case numbers, positivity rates, hospitalizations, or deaths [12,13,14]. Temporal trends in Ct values have demonstrated predictive capabilities, as decreases in median Ct values have been shown to precede confirmed case surges by one to three weeks [15,16,17]. Similarly, they correlate with increased hospitalization rates following a lag of several days to weeks [1,12,14,16,17,18,19,20,21,22]. Such findings reinforce the potential of Ct values as tools for forecasting outbreaks and enabling timely public health responses during pandemics [9,14,15].

A 2023 systematic review demonstrates the exploration of this relationship in multiple studies [23]. For instance, in the UK, a study conducted between 2020 and 2021 observed that declining Ct values preceded increases in SARS-CoV-2 positivity, suggesting their role as early warning indicators of epidemiological changes [24]. Similarly, in the United States, significantly lower Ct values were temporarily associated with future increases in COVID-19 cases during 2020 and 2021 [15]. In South Korea, research spanning February 2020 to January 2022 found that declining Ct values accurately predicted subsequent COVID-19 case surges [12]. In line with these findings, a study in Madrid from 2020 to 2021 utilized time-series analysis to validate Ct values as epidemiological markers for future pandemic waves. The study identified a temporal association between Ct values and both COVID-19 cases and hospital admissions with a lag of several days, concluding that the evolution of average Ct counts at the population level could forecast pandemic dynamics [14].

Further, research conducted in the Czech Republic from April 2020 to April 2022 found that falling median Ct values could signal an imminent epidemic growth, though rising median Ct values did not necessarily preclude it [18]. These findings emphasize the importance of Ct-based monitoring in resource-limited settings, where cost-effective surveillance tools are critical for informing timely public health decisions [9,11,18,19,25,26].

The integration of Ct values into predictive models has provided additional evidence of their utility [23]. Retrospective analyses linking Ct values with community prevalence have informed models that forecast epidemic trends and have suggested a predictive time frame of approximately one to three weeks in daily new cases [15,16,17,19,27,28,29]. More complex models incorporating Ct values have successfully predicted epidemic trajectories, ranging from short-term fluctuations to longer-term incidence patterns [17,19,21,27,28,29,30].

Nevertheless, most studies, including recent reports, have primarily focused on the early stages of the pandemic, particularly not beyond the year 2022 [11,12,14,18]. Consequently, there is limited knowledge regarding the continued association between Ct values at the population level and metrics related to transmission and outcomes in COVID-19 [3,4,11,12,13,14,15,16,17,18,19,20,21,22,23,24,26,27,28,29,30]. This gap is particularly pronounced in the context of emerging variants causing milder disease, changes in human behavior, and increasing hybrid immunity in middle-income countries [31,32,33,34].

The objective of this study was to evaluate the relationships between population-level Ct values from SARS-CoV-2 RT–PCR tests over five years of the pandemic and key COVID-19 transmission and outcome metrics to capture associations both initially and over time. By employing time-series analysis, we aimed to identify temporal associations and lead–lag relationships between Ct values and subsequent trends in case increases, positivity rates, effective reproduction numbers (Rt), hospitalization rates, and mortality. This approach provides evidence of the predictive utility of Ct values in public health decision-making, while also assessing their behavior throughout the pandemic and their role in early warning systems and resource optimization strategies.

## 2. Materials and Methods

### 2.1. Ethics

This study was conducted in accordance with the principles of the Declaration of Helsinki of 1964 and its later amendments, applicable national legislation, and institutional guidelines. The study was evaluated and approved by the Research Committee of the Ministry of Health of Jalisco, and it has been registered in the State Research Registry under the identifier 73/LESP/JAL/2024. Due to the retrospective nature of the study and the exclusive use of de-identified data, informed consent was waived, as approved by the Comité de Ética en Investigación de la Secretaría de Salud de Jalisco (approval number SSJ/DGEICS/DIS/CEI/12/24) and the Comité de Investigación de la Secretaría de Salud de Jalisco (approval number SSJ/DGEICS/DIS/CI/13/24).

### 2.2. Setting

Jalisco, located in western Mexico, covers an area of approximately 78,595 km^2^ and is the third most populous state in the country, with more than 8.8 million inhabitants as estimated in 2024 [35,36]. The state is supported by a healthcare network comprising 88 public hospitals, 5 of which are highly specialized, and 582 health centers across its 125 municipalities. Epidemiological surveillance is organized into 13 health regions covering all municipalities. The State Public Health Laboratory of Jalisco (SPHLJ) serves as a key reference center for diagnostics and surveillance, primarily supporting the uninsured population [37].

### 2.3. Population and Eligibility Criteria

We retrospectively reviewed the registry of the SPHLJ, which included symptomatic patients with influenza-like illness (ILI) who were tested for respiratory viruses from March 2020 to June 2024. A suspected case of ILI was defined as the sudden onset of symptoms accompanied by at least one of the following systemic symptoms: fever or feverishness, cough, or headache. Moreover, at least one of the following localized symptoms was considered: dyspnea, myalgias, arthralgias, odynophagia, chills, chest pain, rhinorrhea, tachypnea, anosmia, dysgeusia, or conjunctivitis [38]. Demographic information, comorbidities, clinical characteristics, and outcomes were systematically extracted from the records. Patients with more than 10% missing data in sociodemographic or clinical fields were excluded from the analysis.

### 2.4. Viral Testing

Nasopharyngeal swabs were collected in viral transport media by trained personnel across the 13 health regions and transported to the SPHLJ under cold chain conditions. Upon arrival, laboratory procedures included viral inactivation, nucleic acid extraction, and viral gene amplification using RT–PCR. Briefly, viral genetic material was extracted via automated platforms that employed magnetic bead-based technology to selectively bind viral RNA, followed by subsequent washing steps for isolation. Two different extraction kits and systems were utilized: the ExiPrep™ Plus Viral DNA/RNA Kit (96 reactions) on the ExiPrep™ 96 platform (Bioneer^®^, Daejeon, Republic of Korea) and the MagNA Pure 96 Small Volume Kit on the MagNA Pure 96 System (Roche^®^, Basel, Switzerland). Viral detection was performed using single and multiplex RT–PCR assays, which target key SARS-CoV-2 genes such as E, N, RdRP, and ORF1ab. The kits used in this study were evaluated and approved by the State Ministry of Health. All kits demonstrated analytical sensitivity of 10 copies/reaction, ensuring reliable detection of viral RNA. The platforms and assays utilized included the COBAS 6800 System (Roche^®^, Basel, Switzerland), Logix Smart RT–PCR Kit (Co-Diagnostics^®^, Salt Lake City, UT, USA), Flu-COVID Vitro Kit (Master Diagnóstica^®^, Madrid, Spain), and the BioFire FilmArray Respiratory Panel (BioFire Diagnostics^®^, Salt Lake City, UT, USA) for the detection of SARS-CoV-2 and other respiratory viruses.

### 2.5. Data Metrics for Time Series Analysis

To compare population-level Ct values with COVID-19 and acute respiratory illness metrics, the weekly median Ct value was calculated from positive RT–PCR results obtained from the SPHLJ. The primary comparison metrics included the weekly number of positive tests (SPHLJ positive SARS-CoV-2 RT–PCR tests) and the positivity rate (SPHLJ SARS-CoV-2 positivity rate). To extend the analysis to state-level transmission and outcome metrics, open-access datasets from the State Ministry of Health and the National Ministry of Health were utilized [39,40,41]. From these datasets, we extracted the following metrics: the number of positive RT–PCR tests statewide (statewide positive SARS-CoV-2 RT–PCR tests), the number of confirmed COVID-19 cases statewide (including those with positive antigen and RT–PCR tests from both public and private institutions), the number of hospitalizations due to acute respiratory illness (including both confirmed and non-confirmed COVID-19 cases), the number of hospitalizations related to COVID-19, the number of deaths due to acute respiratory illness, the number of deaths attributed to COVID-19, and the effective reproduction number (Rt), which was calculated based on the number of confirmed COVID-19 cases at the state level. Additionally, we extracted the predominant SARS-CoV-2 variant (constituting more than 50% of the circulating variants in national sampling during a given period) in Mexico on a monthly basis. This information was subsequently used for time-period analyses according to the predominant variants.

### 2.6. Statistical Analysis

Demographic data were summarized as simple relative frequencies. The percentage of positive test results was calculated by dividing the number of positive tests by the total number of tests conducted within a specified period and expressed as a percentage. The Shapiro–Wilk test was used to evaluate the normality of the data distribution. Proportions were compared using Pearson’s chi-square test or Fisher’s exact test, as appropriate. For quantitative variables, Student’s t test and ANOVA were applied to normally distributed data, whereas the Wilcoxon–Mann–Whitney and Kruskal–Wallis tests were used for nonnormally distributed data.

The effective reproduction number (Rt) was estimated for confirmed COVID-19 cases at the state level to understand the temporal changes in the transmissibility of the virus. To calculate Rt, the incidence at a specific time (I(t)) was divided by the sum of all prior incidences weighted by the probability distribution of the serial interval (w_s), which represents the time between symptom onset in a primary case and a secondary case. The formula used was as follows:

Rt = I(t)/(Σ w_s × I(t − s))


The serial interval was modeled with an average of 5 days and a standard deviation of 2 days, reflecting typical SARS-CoV-2 transmission characteristics. These parameters were chosen to account for the impact of different variants of SARS-CoV-2, including Alpha, Delta, and Omicron, which were prevalent during different periods of the pandemic. The median Rt was derived for each week to provide a robust estimate of the transmission dynamics over the study period.

To evaluate the temporal relationships between SARS-CoV-2 Ct values and key epidemiological metrics of transmissions and outcomes, we employed a series of complementary analytical methods, including cross-correlation functions (CCFs), lagged regression analysis, and vector autoregressive (VAR) modeling. The analysis was segmented into four distinct time periods to account for the evolving dynamics of the pandemic: June 2020–May 2021, June 2021–May 2022, June 2022–May 2023, and June 2023–May 2024. This segmentation allowed for a detailed assessment of changing relationships across different phases of the pandemic, characterized by the emergence of new variants, shifts in public health interventions, and evolving immunity levels.

Additionally, we conducted a segmented analysis based on the predominant circulating variant to capture relationships between Ct values and metrics potentially associated with the behavior of different variants. The initial periods defined were as follows: early lineages (Pango lineages B.1, B.1.1, B.1.1.222, and B.1.1.519) from March 2020 to May 2021; the Alpha variant (WHO-designated variant of concern, Pango lineage B.1.1.7) and the Gamma variant (WHO-designated variant of concern, primarily Pango lineage P.1) co-circulating in June 2021; and the Omicron variant (WHO-designated variant of concern, Pango lineages BA.1, BA.1.1, BA.2, BA.2.12.1, BA.5, and recombinant sublineage XBB.1.5) from December 2021 onwards. However, since we analyzed lags beyond four weeks and due to the short circulation time of the Alpha and Gamma variants, we included them in the group of early lineages.

First, cross-correlation functions (CCFs) were performed to assess the strength and direction of associations between the independent variable (weekly median Ct values) and dependent outcomes, including weekly acute respiratory illness and SARS-CoV-2 metrics such as positivity rates, hospitalizations, and mortality. Lags ranging from 0 to 12 weeks were examined to explore temporal relationships, guided by biological plausibility and previous research [11,12,14,23]. This step identified the optimal time lags where changes in Ct values were most significantly associated with subsequent fluctuations in the dependent variables. The CCF analysis provided critical insights into the lead–lag dynamics of the viral load and epidemiological trends across each pandemic phase.

Following the identification of optimal lags through CCFs, we conducted regression analyses to quantify the temporal associations between Ct values and outcome variables. For each metric, the independent variable (Ct value) lagged according to the CCF findings, ensuring alignment with biologically plausible intervals of viral kinetics and disease progression [23]. Linear regression models were built for each time period, adjusting for covariates such as age, sex, and the presence of comorbidities. These models allowed us to generate effect estimates, providing a detailed understanding of how fluctuations in Ct values influenced key epidemiological outcomes over time.

To further examine temporal causality, we applied Granger causality tests to statistically evaluate whether changes in Ct values could predict subsequent variations in dependent variables, such as hospitalization and mortality rates. This method established the predictive value of Ct values in forecasting key outcomes, identifying periods where they were significant predictors of changes in epidemiological metrics.

Finally, we implemented vector autoregressive (VAR) modeling to examine the interdependencies between Ct values and multiple dependent metrics over time. VAR models, which are particularly suited for multivariate time series analysis, enabled us to model each variable as a function of its own past values and those of other variables in the system. These models were applied across segmented timeframes (annual periods and predominant variant periods) to assess how these relationships evolved with changing pandemic conditions. By incorporating covariates such as age, sex, and comorbidities, we evaluated both unadjusted and adjusted dynamics within the system, providing a comprehensive understanding of the complex interactions between Ct values and epidemiological outcomes.

The statistical analyses were performed using R (version 4.2.1) and Python (version 3.10.4). For R, data import, cleaning, and wrangling were carried out with the readxl, dplyr, and lubridate packages, while ggplot2 was used for graphical visualization. Model fitting, regression diagnostics, and Granger causality tests were conducted with broom and lmtest, and additional vector autoregression (VAR) models were explored using the vars package. Python analyses were conducted in pandas (version 1.4.3) for data manipulation, statsmodels (version 0.13.2) for regression analyses, and VAR models and sklearn (scikit-learn version 1.1.1) for preprocessing and standardization.

## 3. Results

From the RT–PCR test records of the SPHLJ, a total of 127,160 individuals were included, of whom 50,367 tested positive for SARS-CoV-2, and 42,769 had Ct values available for analysis. Similarly, open access datasets documented 659,704 patients at the state level, with 303,020 classified as confirmed COVID-19 cases included in the analysis, as illustrated in Figure 1.

Among the 127,160 individuals tested for SARS-CoV-2 at the SPHLJ, the median age was 38 years (IQR: 27–52), and 54.25% were women (n = 68,983). Of the 50,367 individuals who tested positive, the median age was 42 years (IQR: 30–57), and 52.89% were women (n = 26,624). The distribution of positive cases across the thirteen health regions of the state is illustrated in Figure 2.

Among individuals with positive tests reported at the SPHL, men had a slightly older median age than women (44 [31,32,33,34,35,36,37,38,39,40,41,42,43,44,45,46,47,48,49,50,51,52,53,54,55,56,57,58,59] vs. 41 [29,30,31,32,33,34,35,36,37,38,39,40,41,42,43,44,45,46,47,48,49,50,51,52,53,54,55], *p* < 0.001). Over the five-year period (2020–2024), significant demographic shifts were observed in the number of COVID-19 cases. Women increasingly accounted for a greater proportion of cases, peaking at 62.64% in 2023 (*p* < 0.001). The age distribution revealed a marked decline in cases among individuals aged 18–65 years, decreasing from 82.18% in 2020 to 44.61% in 2024, whereas significant increases were observed in pediatric patients aged 0–2 years and individuals > 65 years, particularly in 2024 (17.65% and 25.98%, respectively; *p* < 0.001). The number of RT–PCR tests performed decreased over the years from 55,806 in 2020 to 2025 in 2024. The remaining age and sex distributions of SARS-CoV-2-positive individuals across years are depicted in Table 1.

The median time from symptom onset to testing was 3 days (IQR: 2–4). Comorbidities were reported in 40.60% of individuals with positive tests (n = 20,454), with hypertension (16.42%), diabetes (13.26%), and obesity (11.36%) being the most prevalent conditions. Asthma and smoking were also documented in 3.10% and 6.16% of patients, respectively. Regarding clinical outcomes, 7430 patients (14.75%) required hospitalization, and pneumonia was diagnosed in 7.05% (n = 3552) of patients at the time of testing. Additionally, 1.62% (n = 816) of patients were reported to be on invasive mechanical ventilation at the time of testing, while COVID-19-related deaths were recorded during follow-up in 7.01% (n = 3531) of patients. Further details on demographic characteristics, comorbidities, and clinical outcomes are provided in Table 2.

Table 3 summarizes the state-level metrics related to acute respiratory disease and COVID-19. During the study period, a total of 659,704 cases of acute respiratory disease were reported, of which 89,410 (13.55%) required hospitalization and 27,303 (4.14%) resulted in deaths. Among these, 303,020 cases (45.93%) were confirmed as COVID-19, with 44,443 (14.67%) requiring hospitalization and 20,325 (6.71%) resulting in COVID-19-related deaths. A significant disparity was observed when stratified by sex, with males exhibiting higher percentages of hospitalizations (18.59% vs. 11.33%, *p* < 0.001) and deaths (9.01% vs. 4.75%, *p* < 0.001) associated with COVID-19. Regarding SARS-CoV-2 testing performed statewide, we observed a progressive reduction in the number of RT–PCR tests during the study period (120,751 tests in 2020 compared to only 2140 tests in 2024). From 2021 onward, rapid antigen tests (RATs) began to predominate over RT–PCR tests, becoming the primary diagnostic tool in subsequent years. The distribution of tests performed at the state level is illustrated in Appendix A.

Among the 42,769 positive SARS-CoV-2 RT–PCR tests analyzed, the median Ct value was 27 (IQR: 22–32). Ct values significantly decreased from 2020 to 2023, indicating increasing viral loads over time, with the lowest median values observed in 2023, which slightly increased in 2024. This pattern was consistent across sexes, with women showing a median Ct of 28 (IQR: 24–33) in 2020, dropping to 22 (IQR: 19–28) in 2023 and slightly increasing to 23 (IQR: 18.48–30.00) in 2024 (*p* < 0.001). A similar trend was observed among men, whose median Ct decreased from 28 (IQR: 23–32) in 2020 to 23 (IQR: 19–30) in 2023, with a slight rebound to 23.94 (IQR: 19.92–31.25) in 2024 (*p* < 0.001). When stratified by age, younger children (0–2 years) presented higher Ct values compared to older children in 2020 and 2021, with median values declining from 29 (IQR: 25–34) in 2020 to 26 (IQR: 19–35) in 2023 and stabilizing at 25.09 (IQR: 20–30.75) in 2024 (*p* = 0.001). In contrast, individuals aged 18–65 years and >65 years exhibited the lowest Ct values in 2023 and 2024, reflecting the highest viral loads, with medians of 22 (IQR: 19–28) and 22 (IQR: 18–26.25), respectively (*p* < 0.001 for both groups). The median Ct values of SARS-CoV-2 RT–PCR-positive tests by year, sex, and age group are presented in Table 4 and further illustrated through density and box plots in Figure 3.

The relationships between Ct values and key epidemiological metrics for acute respiratory disease and COVID-19 are illustrated in Figure 4a–c. During 2020–2021, notable declines in median Ct values were closely followed by increases in confirmed COVID-19 cases, RT–PCR-positive tests, hospitalizations, and deaths, particularly during the late 2020 and early 2021 peaks. A notable drop in Ct values was also observed during the Delta wave (mid-2021) and the Omicron surge (late 2021), corresponding to sharp increases in cases and positive tests. The lowest Ct values were recorded during the Omicron surge (early 2022) and during a fourth wave (dominated by Omicron) in mid-2022, reflecting the high transmissibility and elevated viral loads associated with this variant (Figure 4a). Although hospitalizations and deaths increased following these declines, the magnitude of severe outcomes was reduced likely due to population immunity, vaccination efforts, and the intrinsic characteristics of the Omicron variant, which is widely recognized as less virulent [32]. From 2023 onward, the Ct values stabilized, showing a slight upward trend, which coincided with smaller peaks in cases, positivity rates, and hospitalizations. With respect to positivity rates, strong correlations with declining Ct values were observed from 2020 to 2022, indicating higher viral loads during transmission peaks. From 2023 onward, the positivity rates exhibited smaller peaks and general stabilization (Figure 4b). In relation to the effective reproduction number (Rt), early pandemic surges closely align with reductions in Ct values, highlighting the role of elevated viral loads in driving transmission. However, from 2023 to 2024, this relationship weakened (Figure 4c).

Based on cross-correlation function analysis, significant time-lagged relationships were identified between weekly median Ct values from the SPHLJ and key epidemiological indicators, with notable variations observed across the different study periods. Among the most relevant findings, the strongest correlations for the SPHLJ SARS-CoV-2 RT–PCR positivity rate were observed at lag −1 during the 2020–2021 period (r = −0.48) and at lag −2 in the 2021–2022 period (r = −0.59), indicating that changes in Ct values preceded fluctuations in positivity rates by one to two weeks. Similarly, the positivity rate for all positive RT–PCR tests reported statewide exhibited the strongest correlation at lag −2 from 2021 to 2022 (r = −0.58), suggesting a two-week delay. Notably, the basic reproduction number (Rt) exhibited the strongest correlation at lag −6 from 2021 to 2022 (r = −0.44), highlighting a longer lag period indicative of a delayed but significant association between Ct dynamics and viral transmissibility. For COVID-19-related hospitalizations, the correlation was most pronounced at lag −4 from 2023 to 2024 (r = −0.42), whereas for COVID-19-related mortality, the strongest correlation occurred at lag −5 during the same period (r = −0.35). The cross-correlations between Ct values and transmission metrics, as well as clinical outcomes for COVID-19 and acute viral respiratory illness, are presented in Figure 5 and Appendix A.

Overall, there was a noticeable decrease in the strength of negative correlations between Ct values and key epidemiological metrics. During the earlier period of the pandemic (2020–2021), stronger negative correlations were observed at negative lags, likely reflecting the closer association between higher viral loads (lower Ct values) and adverse epidemiological outcomes during this phase. In contrast, in more recent periods (2022–2023 and 2023–2024), these correlations weakened or even disappeared.

In the cross-correlation function analyses of weekly median Ct values from the SPHLJ with COVID-19 metrics across distinct time periods defined by the predominance of different SARS-CoV-2 variants, varying strengths of negative correlations were observed. During the period dominated by initial lineages, few significant negative correlations were identified at negative lags, except for the reproduction number (Rt), which exhibited a strong negative correlation at a lag of −6 weeks (r = −0.48).

In the Delta-dominant period, significant negative correlations were observed across multiple metrics. For instance, there was a negative correlation between Ct values and the SPHLJ SARS-CoV-2 positivity rate (r = −0.49 at a lag of −6 weeks), while COVID-19-related hospitalizations showed a correlation of r = −0.39 at a lag of −2 weeks. Similarly, the reproduction number (Rt) demonstrated a strong negative correlation (r = −0.39) at a lag of −3 weeks, and the statewide SARS-CoV-2 RT–PCR positivity rate exhibited a negative correlation at a lag of −5 weeks (r = −0.42). These findings highlight the predictive capacity of Ct values during this phase.

In the Omicron-dominant period, although correlations were generally weaker, some metrics continued to reflect the expected trends. For example, the correlation between Ct values and COVID-19-related hospitalizations was r = −0.08 at a lag of −1 week. Notably, the statewide SARS-CoV-2 RT–PCR positivity rate (r = −0.28 at a lag of −1 week) and the statewide SARS-CoV-2 test’s positivity rate (r = −0.35 at a lag of −1 week) demonstrated significant correlations. These findings align with the hypothesis that Ct values can act as early indicators of subsequent reductions in disease burden.

Overall, the analysis indicates a progressive decline in the strength of correlations between Ct values and adverse metrics as the pandemic evolved. The Delta period demonstrated the strongest and most consistent negative correlations, underscoring its predictive utility for metrics like hospitalizations and deaths at intermediate lags (−2 to −6 weeks). In contrast, the Omicron period, characterized by weaker correlations at shorter lags, aligns with the reduced severity and faster transmission dynamics of the Omicron variant. The cross-correlations between Ct values and transmission metrics, as well as clinical outcomes for COVID-19 and acute viral respiratory illness, are presented in Figure 6 and Appendix A.

The lagged regression analysis, adjusted for covariates, revealed a pattern of significant negative associations between weekly median Ct values from the SPHLJ and various epidemiological metrics across distinct pandemic periods. For instance, concerning the SPHLJ SARS-CoV-2 RT–PCR positivity rate, a strong inverse relationship emerged from 2020 to 2021 at lag −1 (estimate: −3.23; 95% CI: −4.72 to −1.73; *p* < 0.001), suggesting that decreases in Ct values anticipated increases in positivity rates roughly one week later. Similarly, a more attenuated effect persisted into 2023–2024 at lag −6 (estimate: −0.45; 95% CI: −0.78 to −0.12; *p* = 0.009), indicating that the predictive lead time varied over the course of the pandemic. This trend was also evident for confirmed COVID-19 cases, with a strong negative association observed in 2020–2021 at lag −1 (estimate: −256.80; 95% CI: −414.51 to −99.10; *p* = 0.002) and a continued, albeit weaker, significant effect in 2022–2023 (estimate: −65.53; 95% CI: −127.95 to −3.11; *p* = 0.040). In addition, the statewide RT–PCR positivity rate showed an inverse correlation from 2021 to 2022 at lag −2 (estimate: −1.28; 95% CI: −1.99 to −0.57; *p* < 0.001), underscoring the consistently predictive nature of Ct values for key testing metrics. Notably, the effective reproduction number (Rt) also demonstrated a significant inverse association, although with a lower magnitude, at lag −6 during the 2021–2022 period (estimate: −0.041; 95% CI: −0.07 to −0.012; *p* = 0.007).

Moreover, clinical outcomes were similarly affected. For COVID-19-related hospitalizations, a significant negative association was identified from 2020 to 2021 at lag −1 (estimate: −62.05; 95% CI: −117.18–−6.92; *p* = 0.028). Similarly, COVID-19-related deaths were negatively associated from 2020 to 2021 at lag −1 (estimate: −36.63; 95% CI: −66.34 to −6.91; *p* = 0.017) and persisted in 2023–2024 at lag −5 (estimate: −0.145; 95% CI: −0.265 to −0.025; *p* = 0.019).

In summary, the lagged regression analysis with respect to annual periods demonstrated a progressive weakening of negative associations between Ct values and key epidemiological metrics. Early in the pandemic (2020–2021), strong and statistically significant correlations were observed at shorter lags, reflecting the closer link between higher viral loads and adverse outcomes. However, in later periods (2022–2024), these associations diminished significantly. The complete set of regression results, segmented by year, is provided in Figure 7 and Appendix A.

The lagged regression analysis, adjusted for covariates, of weekly median Ct values from the SPHLJ and COVID-19 transmission and outcome metrics revealed significant relationships during periods of predominant variant circulation. During the initial lineages and alpha and gamma variants period, few metrics exhibited statistically significant associations. For example, the effective reproduction number (Rt) showed a negative association with Ct values at lag −6 weeks (estimate = −0.189, 95% CI: −0.234 to −0.144, *p* < 0.001). Other metrics did not achieve significance.

During the Delta-dominant period, stronger and more consistent negative associations were observed across key metrics. COVID-19-related hospitalizations were significantly associated with Ct values at lag −2 weeks (estimate = −94.62, 95% CI: −188.34 to −0.89, *p* = 0.048), while the SPHLJ SARS-CoV-2 positivity rate exhibited a significant negative association at lag −6 weeks (estimate = −4.89, 95% CI: −8.46 to −1.32, *p* = 0.01). Additionally, the statewide SARS-CoV-2 RT–PCR positivity rate was significantly associated at a lag of −5 weeks (estimate = −3.48, 95% CI: −6.69 to −0.28, *p* = 0.03), and the statewide SARS-CoV-2 tests positivity rate (which includes both RT–PCR and antigen tests) showed a significant association at a lag of −2 weeks (estimate = −2.72, 95% CI: −5.37 to −0.06, *p* = 0.04). Regarding other clinical outcomes, acute respiratory disease-related deaths (estimate = −57.25, 95% CI: −107.907 to −6.60, *p* = 0.03) and COVID-19-related deaths (estimate = −50.45, 95% CI: −95.80 to −5.11, *p* = 0.03) both demonstrated significant negative associations at lag −5 weeks.

During the Omicron-dominant period, significant associations were weaker and less frequent, likely reflecting the reduced severity and rapid transmission dynamics of this variant. For instance, the statewide SARS-CoV-2 test’s positivity rate was significantly associated with Ct values at lag −1 week (estimate: −1.51; 95% CI: −2.20 to −0.81; *p* < 0.01). Other weaker associations included the SPHLJ SARS-CoV-2 positivity rate at lag −2 weeks (estimate: −0.72; 95% CI: −1.41 to −0.02; *p* = 0.04), the statewide SARS-CoV-2 RT–PCR positivity rate at lag −1 week (estimate: −0.97; 95% CI: −1.56 to −0.38; *p* = 0.01), the statewide SARS-CoV-2 test’s positivity rate at lag −1 week (estimate: −1.51; 95% CI: −2.20 to −0.81; *p* < 0.001), and the effective reproduction number (Rt) at lag −4 weeks (estimate: −0.03; 95% CI: −0.05 to −0.01; *p* = 0.01). The complete sets of regression results, segmented by periods of predominant variant circulation, are presented in Figure 8 and Appendix A.

The analysis combining Granger causality and vector autoregression (VAR) models revealed significant relationships between Ct values and key epidemiological metrics during specific periods. For the SPHLJ positive SARS-CoV-2 RT–PCR tests, the period from June 2020 to May 2021 showed a highly significant Granger causality association (*p* = 0.0008), indicating that Ct values effectively predicted changes in this metric. This was further supported by the VAR model, which demonstrated a substantial negative coefficient (−47.62; *p* = 0.0011) and a high adjusted R^2^ of 0.92, highlighting a strong and consistent temporal association.

For the SPHLJ SARS-CoV-2 RT–PCR positivity rate, significant relationships were observed across two distinct periods. From June 2020 to May 2021, Granger causality (*p* = 0.026) and VAR analysis (coefficient: −2.02; *p* = 0.0083) confirmed the predictive value of Ct values, with an adjusted R^2^ of 0.85. Similarly, from June 2021 to May 2022, Granger causality remained significant (*p* = 0.0051), while the VAR analysis (coefficient: −0.71; *p* = 0.047) demonstrated a smaller effect size compared to the earlier period.

For the effective reproduction number (Rt), the period from June 2021 to May 2022 exhibited significant associations in both Granger causality (*p* = 0.039) and VAR analysis (coefficient: −0.012; *p* = 0.025). The adjusted R^2^ of 0.98 underscored the robustness of the model, indicating a strong temporal relationship between Ct values and viral transmissibility.

During the period dominated by the initial lineages, as well as the Alpha and Gamma variants, significant associations were observed for Rt. At an optimal lag of 4 weeks, Ct values demonstrated a strong negative association with Rt (coefficient: −0.24; *p* < 0.001; R^2^ = 0.47), underscoring their predictive utility in estimating viral transmissibility during this phase.

In the Delta-dominant period, the SPHLJ SARS-CoV-2 RT–PCR positivity rate displayed a significant association with Ct values at an optimal lag of 1 week, as indicated via the Granger causality test (*p* = 0.009). During the Omicron-dominant period, significant associations were primarily identified for statewide metrics. VAR analysis showed that the statewide SARS-CoV-2 RT–PCR positivity rate exhibited a strong negative association at an optimal lag of 1 week (coefficient = −0.559, *p* = 0.004, R^2^ = 0.64). The statewide SARS-CoV-2 positivity rate also showed a significant association at the same lag (coefficient = −0.97, *p* < 0.001, R^2^ = 0.62), although these metrics were not significant in the Granger test. The summary results of the Granger causality tests and VAR analysis are presented in Table 5. An extended version, detailing all metrics analyzed by year, is available in Appendix A, while results segmented by periods of variant predominance are provided in Appendix A.

## 4. Discussion

Our study provides a comprehensive analysis of the relationships between population-level SARS-CoV-2 Ct values and key epidemiological metrics, including positivity rates, reproduction numbers (Rt), hospitalizations, and mortality. Notably, we identified significant temporal associations between declining Ct values and subsequent increases in these metrics, particularly during the early years of the pandemic, underscoring their predictive utility in similar future scenarios.

Our analyses revealed that the predictive value of median weekly Ct values for changes in COVID-19 transmission and outcome metrics was more pronounced during the early pandemic years, progressively weakening over time. Cross-correlation functions identified stronger negative correlations during 2020–2021 and 2021–2022, with optimal lags occurring earlier in these periods and shifting to later time points in subsequent years. These trends were particularly evident during the Delta-dominant period, which demonstrated the most robust and consistent negative associations for metrics like hospitalizations and deaths at intermediate lags (−2 to −6 weeks). In contrast, the Omicron-dominant period showed weaker and less frequent associations, likely due to the reduced severity and rapid transmission dynamics of this variant.

Lagged regression analyses, adjusted for covariates such as age, sex, and comorbidities, reinforced these observations. The magnitude and significance of Ct value effects on epidemiological metrics were notably greater in the early pandemic phases, aligning with the periods of higher viral loads and more severe outcomes. Similarly, vector autoregressive models highlighted significant influences of Ct values on weekly epidemiological metrics during the initial stages of the pandemic but not in later periods. Granger causality tests further validated the temporal directionality of these associations, confirming that Ct values served as early indicators of epidemiological trends during the initial phases of the pandemic.

The weakening of these associations during the later stages of the pandemic can be attributed to multiple factors, including widespread vaccination campaigns, increased population-level immunity from prior infections and vaccination, and the emergence of new viral variants with differing transmissibility and virulence [31,34,42,43,44,45,46,47]. Additionally, the transition to RATs during the Omicron surge was necessitated by the high transmissibility of the variant and logistical challenges in maintaining RT–PCR testing capacity [42,48]. While RATs are less sensitive than RT–PCR, they offer quick results and ease of use, enabling timely isolation measures and reducing the burden on healthcare systems; however, this reduced sensitivity of RATs, particularly in cases with low viral loads, may lead to underreporting of cases, thereby affecting the accuracy of case counts and transmission metrics [49].

Behavioral adaptations, including changes in testing strategies and the relaxation of non-pharmaceutical interventions, likely contributed to these shifts [42,50,51]. These adaptations not only limit the availability of Ct value data but also affect the accuracy of key epidemiological metrics such as the time-varying effective reproductive number (Rt) [33,50]. When shifts from RT–PCR to RATs occur and underreporting is present, the fraction of detected and reported cases becomes inconsistent over time, potentially biasing the estimates of Rt [50]. The emergence of Omicron has also influenced population immunity dynamics. In the United States, for example, protection against Omicron infection and severe disease increased significantly between December 2021 and November 2022, driven by both natural infections and vaccination efforts [52]. This heightened immunity, combined with the inherent characteristics of the Omicron variant, which is associated with milder disease outcomes, has contributed to a lower case fatality rate compared to earlier variants [31,53]. Consequently, these transitions may have diminished the utility of Ct values as proxies for understanding viral dynamics and epidemiological trends. This highlights the importance of contextualizing Ct value analyses within specific epidemiological and temporal frameworks to accurately interpret their implications for public health strategies.

Our analysis also revealed significant variation in Ct values across demographic groups, with notable differences according to age and sex. Lower Ct values—reflecting higher viral loads—were observed in older age groups and male patients, which aligns with the increased severity of disease and worse outcomes reported in these populations [4,9,54,55,56]. However, although older patients were reported to have significantly higher viral loads compared to younger individuals during the early stages of the pandemic, this difference diminished over time, likely influenced by public health measures and evolving viral transmission dynamics [57].

The strengths of this study lie in the extensive dataset utilized and the robust analytical approach applied. By employing cross-correlation functions, lagged regression models, and vector autoregressive analyses, we captured complex temporal relationships and provided statistically rigorous estimates of the predictive value of Ct metrics. Additionally, the segmentation of data into distinct time periods allowed us to account for evolving pandemic dynamics with the emergence of new SARS-CoV-2 variants. Unlike prior studies that primarily focused on the initial associations between Ct values from RT–PCR-positive tests and epidemiological metrics during the early stages of the pandemic, our study, to the best of our knowledge, is the first to comprehensively examine both the early associations and the observed effects during the later phases of the pandemic in a middle-income Latin American country [1,15,16,17,19,20,21,22,26,28,29,30,58,59,60,61].

However, certain limitations must be acknowledged. The retrospective nature of the study introduces potential biases related to data completeness and accuracy. While Ct values hold promise as epidemiological tools, several methodological considerations must be addressed. Variability in Ct values can arise from differences in sampling techniques, the timing of specimen collection relative to symptom onset, assay protocols, target gene selection, and laboratory platforms [3,62]. Although the diagnostic platforms described in our state were approved by the National Ministry of Health, ensuring a detection limit of 10 copies/reaction, multiple platforms were ultimately used, which could introduce variability in the results. In this context, regarding the statewide RT–PCR tests, these included multiple additional laboratories, in addition to the SPHLJ, that utilized diagnostic platforms (also approved by the Ministry of Health), for which we do not have precise information about the specific kits employed by each laboratory. The standardization of testing protocols and the normalization of Ct values across platforms are critical to ensure their reliability and comparability [63]. Additionally, the reduction in the number of RT–PCR tests performed at our reference laboratory was accompanied by a progressively greater reliance on rapid antigen tests (RATs), which cannot be quantified with complete certainty across the state (Appendix A). This shift may have contributed to the underreporting of cases and a lower number of RT–PCR tests with available Ct values, potentially impacting the predictive utility of Ct values for epidemiological metrics. Finally, one aspect that could have strengthened our analysis would have been the quantification of vaccination coverage (beyond the stages of sequential rollout in the state), as this may have influenced the predictive capacity of Ct values and provided valuable information in this regard.

The implications of our findings extend beyond the immediate context of SARS-CoV-2. Ct values, as scalable and readily available metrics, have the potential to transform public health responses to other respiratory viruses. By integrating Ct-based monitoring into global surveillance frameworks, we can increase preparedness for future pandemics. For instance, combining Ct trends with genomic sequencing data could provide a more comprehensive understanding of viral evolution and transmissibility.

Our findings highlight the importance of interdisciplinary collaboration to enhance the utility of Ct metrics. By fostering partnerships among epidemiologists, data scientists, and public health practitioners, advanced predictive models can be developed using machine learning and artificial intelligence to improve outbreak forecasting and optimize resource allocation. Future research should build on these findings by integrating Ct values with other epidemiological and clinical datasets. For instance, combining Ct trends with vaccination coverage rates or genomic surveillance data could offer deeper insights into the factors influencing transmission dynamics. Additionally, further studies are required to validate the applicability of Ct-based monitoring in diverse settings, populations, and pandemic timeframes, with a particular emphasis on low- and middle-income countries.

## 5. Conclusions

In conclusion, our study highlights the predictive value of population-level SARS-CoV-2 Ct values in monitoring COVID-19 transmission dynamics and clinical outcomes. We found that declining median Ct values, indicative of increased viral loads, preceded significant surges in positivity rates, effective reproduction numbers, hospitalizations, and mortality, particularly during the early phases of the pandemic. Temporal shifts in these associations, observed across distinct phases of the pandemic, reflect the evolving interplay between viral transmission dynamics, population immunity, and public health interventions. The weakening of these associations in later stages can be attributed to increased immunity, the emergence of less severe variants, and transitions in diagnostic strategies. Nonetheless, integrating Ct values into surveillance systems during early outbreak phases remains critical for early warning and resource allocation. Future research combining Ct trends with genomic and vaccination data could further enhance predictive models, enabling robust and informed responses to respiratory viral pandemics.

## Figures and Tables

**Figure 1 viruses-17-00103-f001:**
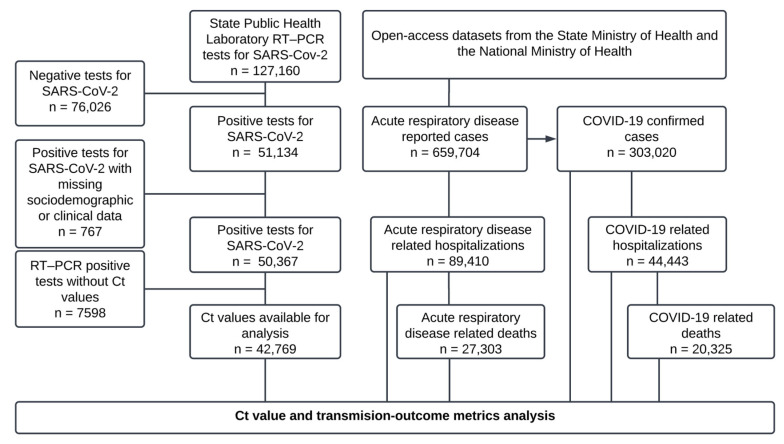
Flowchart of case and test selection.

**Figure 2 viruses-17-00103-f002:**
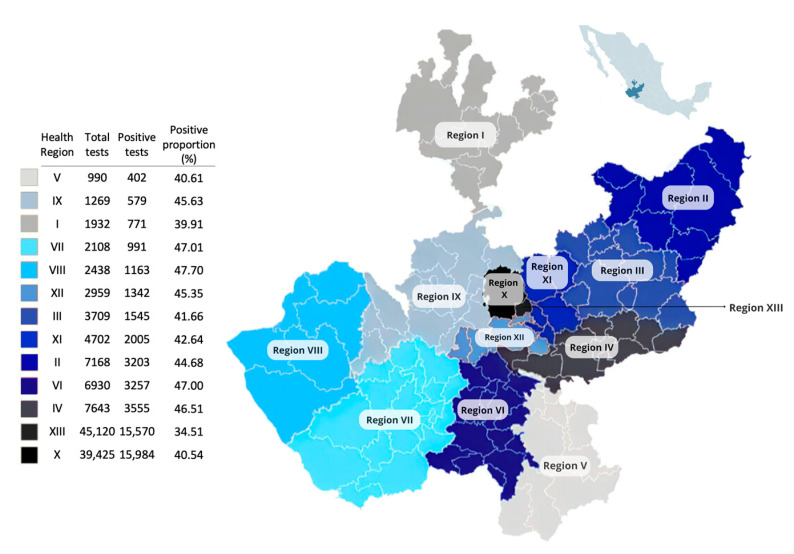
Geographic distribution of RT–PCR testing for SARS-CoV-2 and the proportion of positive tests across the 13 health regions in Jalisco, Mexico.

**Figure 3 viruses-17-00103-f003:**
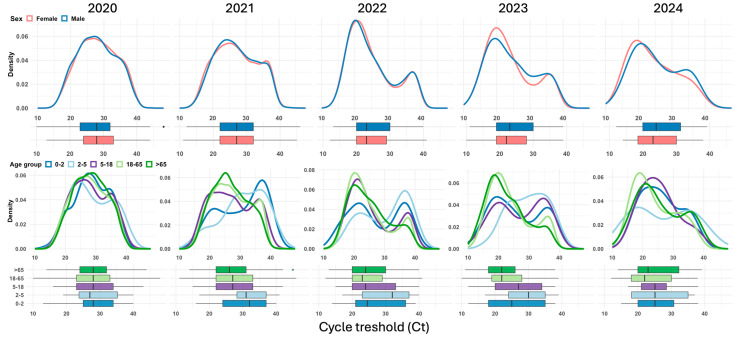
Annual distribution of SARS-CoV-2 cycle threshold (Ct) values from 2020 to 2024, stratified by sex (top row: density plots and box plots) and age group (bottom row: density plots and box plots). The *x*-axis represents the Ct value in both the density and box plots, while the *y*-axis represents the density for each subgroup (sex or age group).

**Figure 4 viruses-17-00103-f004:**
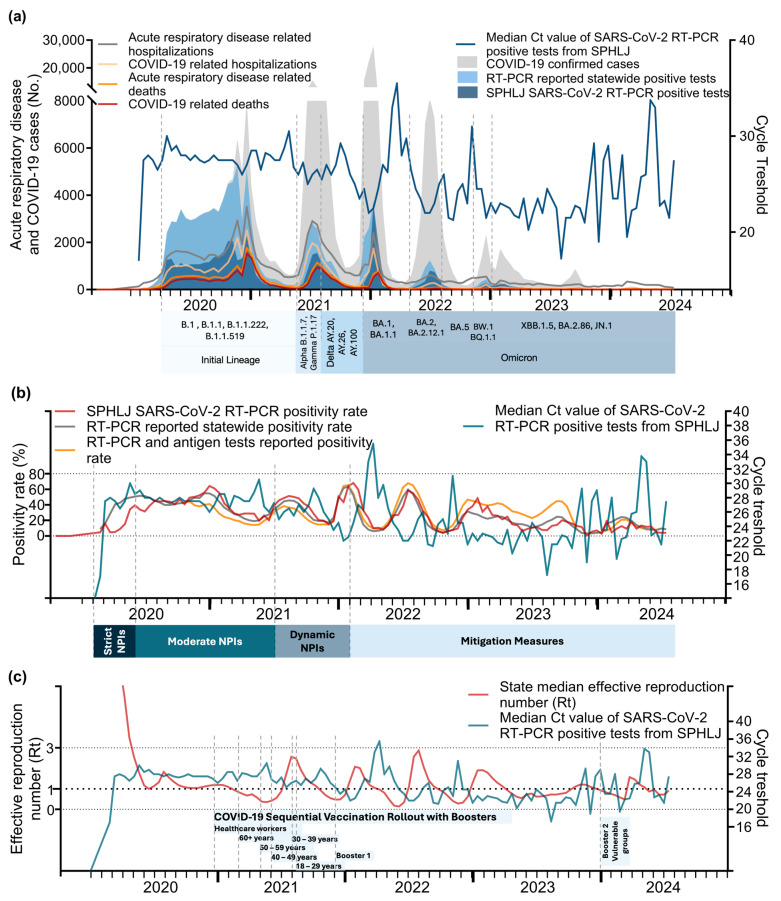
Median Ct values from the SPHLJ positive SARS-CoV-2 RT–PCR tests and their relationship with acute respiratory disease and COVID–19 metrics: (**a**) median Ct values alongside positive tests, confirmed cases, and outcomes; (**b**) median Ct values alongside the SPHLJ SARS-CoV-2 RT–PCR positivity rate, statewide SARS-CoV-2 test positivity rate, and statewide SARS-CoV-2 test positivity rate from RT–PCR and antigen tests; (**c**) median Ct values alongside the effective reproduction number (Rt). Ct: cycle threshold; Rt: effective reproduction number; SPHLJ: State Public Health Laboratory of Jalisco.

**Figure 5 viruses-17-00103-f005:**
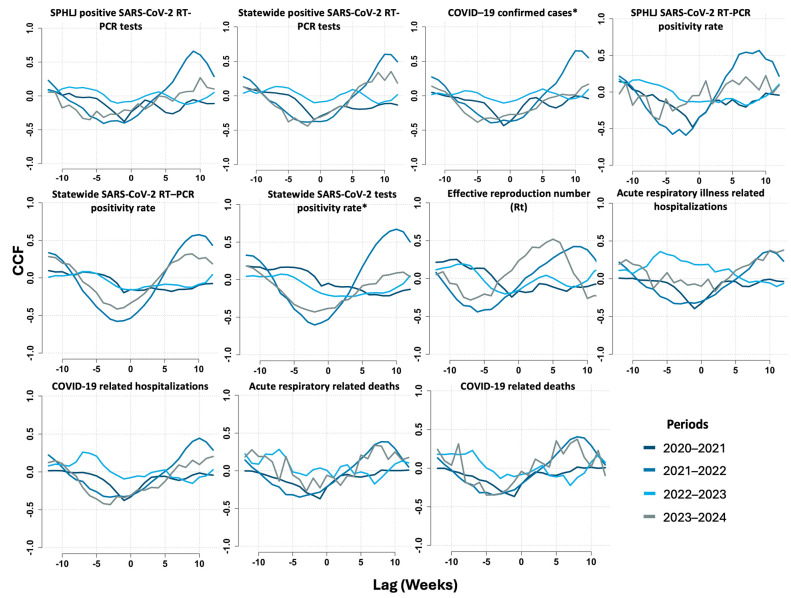
Cross-correlation analysis of COVID-19 metrics and SARS-CoV-2 Ct lag values across the studied time periods. The *x*-axis represents lag in weeks, and the *y*-axis shows CCF values, with variations indicating time-dependent correlations for each metric. * Includes positive RT–PCR and rapid antigen tests.

**Figure 6 viruses-17-00103-f006:**
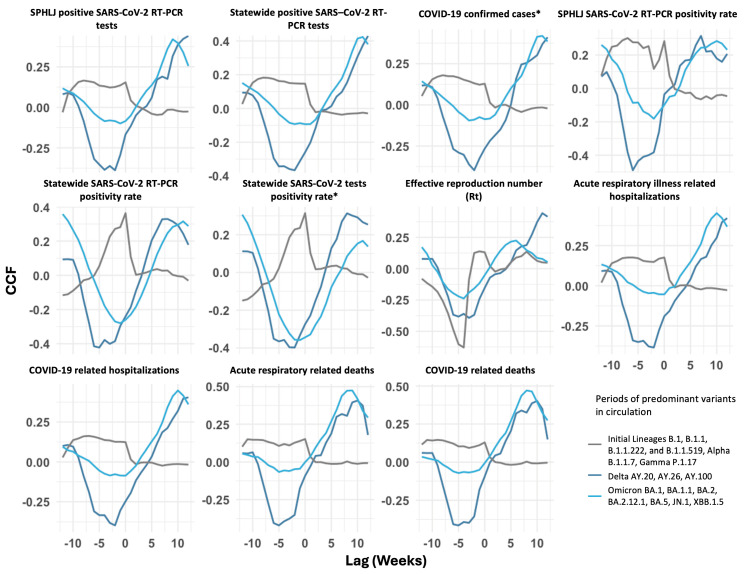
Cross-correlation analysis of COVID-19 metrics and SARS-CoV-2 Ct lag values during periods of predominant variant circulation. The *x*-axis shows Ct lag in days, and the *y*-axis shows CCF values with variations indicating time-dependent correlations for each metric. * Includes positive RT–PCR and rapid antigen tests.

**Figure 7 viruses-17-00103-f007:**
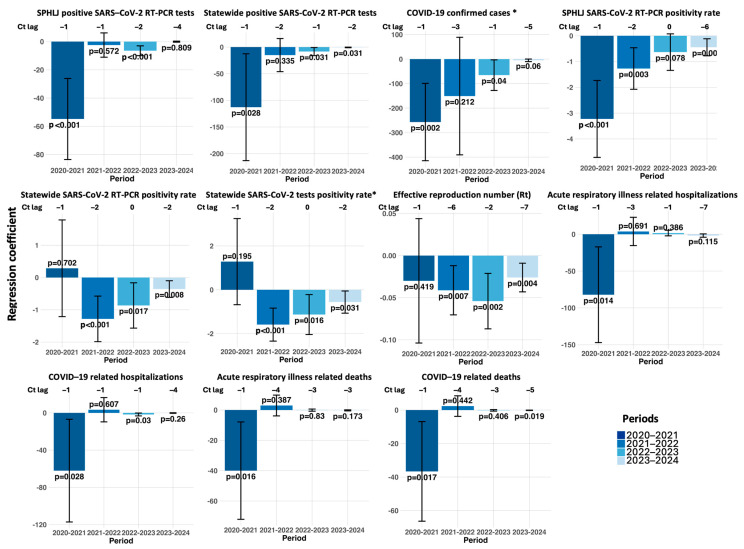
Lagged regression analysis of SARS-CoV-2 Ct values and COVID-19 transmission and outcome metrics. The *x*-axis shows Ct lag in days, and the *y*-axis shows CCF values with 95% confidence intervals. Significant *p*-values (*p* < 0.05) are indicated. Includes RT–PCR and rapid antigen tests. * Includes positive RT–PCR and rapid antigen tests.

**Figure 8 viruses-17-00103-f008:**
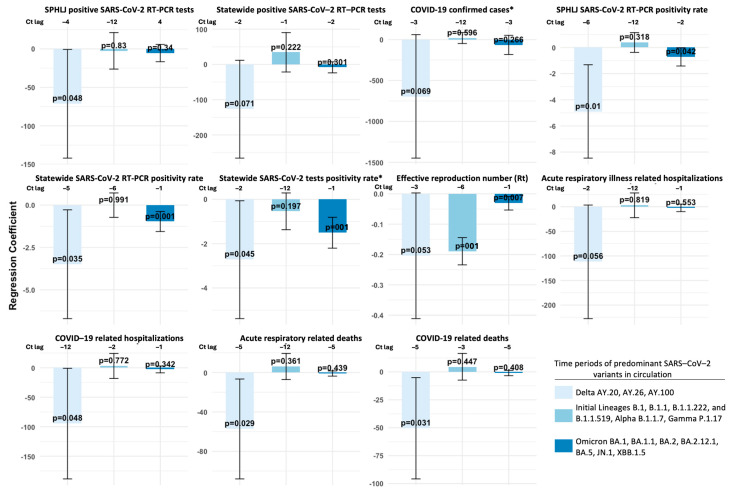
Lagged regression analysis of SARS-CoV-2 Ct lag and COVID-19 transmission and outcome metrics during periods of predominant variant circulation. * Includes positive RT–PCR and rapid antigen tests.

**Table 1 viruses-17-00103-t001:** Distribution by age and sex of individuals with RT–PCR-positive tests reported at the SPHLJ and proportions of positive tests during the study period.

	2020		2021		2022		2023		2024		
	n	%	n	%	n	%	n	%	n	%	*p*-Value
Sex											
Women	10,853	50.09	8062	51.20	6563	58.96	1043	62.64	103	50.49	<0.001
Men	10,814	49.91	7638	48.80	4568	41.04	622	37.36	101	49.51	
Age group (years)											
0–2	313	1.44	115	0.73	175	1.57	100	6.01	36	17.65	<0.001
2–5	43	0.20	38	0.24	72	0.65	28	1.68	6	2.94	
5–18	585	2.70	700	4.46	709	6.37	77	4.62	18	8.82	
18–65	17,806	82.18	11,956	76.15	9065	81.44	1243	74.65	91	44.61	
>65	2920	13.48	2891	18.41	1110	9.97	217	13.03	53	25.98	
Total positive tests and proportion of positive tests	21,667	38.83 *	15,700	44.71 *	11,131	44.29 *	1665	20.02 *	204	10.07 *	
Number of tests performed	55,806		35,114		25,133		8315		2025		

* Proportions were calculated based on the total number of tests performed.

**Table 2 viruses-17-00103-t002:** Demographic, comorbidity, and clinical characteristics of SPHL SARS-CoV-2-positive patients according to sex.

Variable	Total Positive Tests (n = 50,367)	Positive Tests in Women (n = 26,624)	Positive Tests in Men (n = 23,743)	*p* Value
Demographic variables				
Age (years)—median, (IQR)	42 (30—57)	41 (29–55)	44 (31–59)	<0.001
Metropolitan areas n (%)	34,909 (69.31)	18,626 (69.96)	16,283 (68.58)	<0.001
Nonmetropolitan areas n (%)	15,458 (30.69)	7998 (30.04)	7460 (31.42)	<0.001
Comorbidities—n (%)	20,454 (40.60)	10,569 (39.69)	9885 (41.63)	<0.001
Asthma	1566 (3.10)	1045 (3.93)	521 (2.19)	<0.001
Diabetes mellitus	6683 (13.26)	3295 (12.38)	3388 (14.27)	0.111
Cardiovascular disease	1085 (2.15)	520 (1.95)	565 (2.38)	0.059
COPD diagnosis	912 (1.81)	438 (1.65)	474 (2.00)	0.101
Chronic kidney disease	851 (1.69)	381 (1.43)	470 (1.98)	<0.001
Hypertension	8271 (16.42)	4201 (15.78)	4070 (17.14)	0.043
Immunosuppression	741 (1.47)	375 (1.41)	366 (1.54)	0.678
Obesity	5724 (11.36)	3040 (11.42)	2684 (11.30)	<0.001
Smoking	3102 (6.16)	1074 (4.03)	2028 (8.54)	<0.001
HIV infection	201 (0.40)	67 (0.25)	134 (0.56)	<0.001
Other comorbidities	2774 (5.51)	1682 (6.32)	1092 (4.60)	<0.001
Clinical and outcome variables				
Days from symptom onset to testing—median (IQR)	3 (2–4)	2 (2–4)	3 (2–4)	<0.001
RT–PCR Ct value—median, (IQR)	27 (22–32)	27 (22–32)	27 (22–32)	0.022
Pregnancy—n (%)	146 (0.55)	146 (0.55)	-	-
Hospitalization—n (%)	7430 (14.75)	3016 (11.33)	4414 (18.59)	<0.001
Pneumonia diagnosis—n (%)	3552 (7.05)	1392 (5.23)	2160 (9.10)	<0.001
Invasive mechanical ventilation—n (%)	816 (1.62)	293 (1.10)	523 (2.20)	<0.001
COVID-19-related death—n (%)	3531 (7.01)	1392 (5.23)	2139 (9.01)	<0.001

SPHL: State Public Health Laboratory of Jalisco.

**Table 3 viruses-17-00103-t003:** Acute respiratory disease and SARS-CoV-2 statewide metrics.

Variable—n (%)	Total Cases of Acute Respiratory Disease (n = 659,704)	Cases of Acute Respiratory Disease in Women (n = 366,348)	Cases of Acute Respiratory Disease in Men (n = 293,356)	*p* Value
Acute-respiratory-disease-related hospitalizations	89,410 (13.55)	40,339 (11.01)	49,071 (16.73)	<0.001
Acute-respiratory-disease-related deaths	27,303 (4.14)	10,683 (2.92)	16,620 (5.67)	<0.001
COVID-19-confirmed cases	303,020 (45.93)	163,872 (44.73)	139,148 (47.43)	<0.001
**Variable—n (%)**	**Total cases of acute respiratory disease** **(n = 303,020)**	**Cases of acute respiratory disease in women** **(n = 163,872)**	**Cases of acute respiratory disease in men** **(n = 139,148)**	***p* value**
COVID-19-related hospitalizations	44,443 (14.67)	18,573 (11.33)	25,870 (18.59)	<0.001
COVID-19-related deaths	20,325 (6.71)	7784 (4.75)	12,541 (9.01)	<0.001

**Table 4 viruses-17-00103-t004:** Median Ct values for SARS-CoV-2 RT–PCR tests by year, sex, and age group.

	2020	2021	2022	2023	2024	*p* Value *
Median Ct value	28 (24–33)	27 (22.0–32.0)	23 (20.0–29.0)	22 (19.0–29.0)	23 (19.0–30.08)	<0.001
Sex						
Women	28.0 (24.0–33.0)	27.0 (22.0–32.0)	23.0 (20.0–29.0)	22.0 (19.0–28.0)	23.0 (18.5–30.0)	<0.001
Men	28.0 (23.0–32.0)	27.0 (22.0–32.0)	23.0 (20.0–30.0)	23.0 (19.0–30.0)	23.9 (19.9–31.3)	<0.001
Age group						
0–2	29.0 (25.0–34.0)	33.0 (24.0–37.0)	27.5 (21.0–37.0)	26.0 (19.0–35.0)	25.1 (20.0–30.8)	0.001
2–5	27.0 (24.0–34.5)	31.0 (26.5–37.0)	29.5 (23.0–37.0)	30.0 (23.0–34.0)	25.1 (18.1–35.0)	0.659
5–18	28.0 (24.0–33.0)	27.0 (22.0–33.0)	23.0 (20.0–32.0)	27.0 (20.0–34.0)	24.0 (20.0–28.0)	<0.001
18–65	28.0 (24.0–33.0)	27.0 (22.0–33.0)	23.0 (20.0–29.0)	22.0 (19.0–28.0)	22.0 (18.0–30.1)	<0.001
>65	28.0 (24.0–32.0)	26.0 (22.0–31.0)	24.0 (20.0–30.0)	22.0 (18.0–26.25)	23.1 (20.0–33.0)	<0.001

* Differences calculated within the same group across years.

**Table 5 viruses-17-00103-t005:** Summary of the Granger causality test and vector autoregression model results.

Period	Metric	Optimal Lag	Granger Test *p* Value	Ct Lag Coefficient	Ct Lag *p*-Value	Adjusted R^2^
2020–2021	SPHLJ positive SARS-CoV-2 RT–PCR tests	1	0.001	−47.624	0.001	0.927
	SPHLJ SARS-CoV-2 positivity rate	2	0.026	−2.025	0.008	0.853
	Statewide SARS-CoV-2 positivity rate *	2	0.024	−0.012	0.068	0.980
	Acute-respiratory-disease-related deaths	3	0.114	−20.153	0.018	0.941
	COVID-19-related deaths	3	0.071	−19.426	0.011	0.944
2021–2022	SPHLJ SARS-CoV-2 positivity rate	2	0.005	−0.709	0.047	0.826
	Effective reproduction number (Rt)	3	0.039	−0.012	0.025	0.980
2022–2023	SPHLJ positive SARS-CoV-2 RT–PCR tests	2	0.015	−1.627	0.079	0.942
Initial Lineages and Alpha and Gamma variants	SPHLJ SARS-CoV-2 positivity rate	3	0.006	0.457	0.208	0.525
	Statewide SARS-CoV-2 RT–PCR positivity rate	1	0.029	0.305	0.107	0.799
	Statewide SARS-CoV-2 positivity rate *	1	0.009	0.391	0.102	0.720
	Effective reproduction number (Rt)	4	<0.001	−0.238	<0.001	0.473
Delta variant	SPHLJ SARS-CoV-2 RT–PCR positivity rate	1	0.009	−0.399	0.694	0.773
Omicron variants	Statewide SARS-CoV-2 RT–PCR positivity rate	1	0.289	−0.559	0.004	0.643
	Statewide SARS-CoV-2 positivity rate *	1	0.399	−0.970	<0.001	0.619

Ct: Cycle threshold; Rt: effective reproduction number; SPHLJ: State Public Health Laboratory of Jalisco. * Includes positive RT–PCR and rapid antigen tests. Pangolin lineages corresponding to WHO variants: Initial Lineages B.1, B.1.1, B.1.1.222, and B.1.1.519; Alpha variant B.1.1.7; Gamma variant P.1.17; Delta variants AY.20, AY.26, and AY.100; and Omicron variants BA.1, BA.1.1, BA.2, BA.2.12.1, BA.5, JN.1, and XBB.1.5.

## Data Availability

The raw data supporting the conclusions of this article will be made available by the authors upon request.

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
