# Peer review of "Population-Level SARS-CoV-2 RT–PCR Cycle Threshold Values and Their Relationships with COVID-19 Transmission and Outcome Metrics: A Time Series Analysis Across Pandemic Years"

_viruses, 2025, doi:10.3390/v17010103_

Round 1

Reviewer 1 Report

Comments and Suggestions for Authors

Conducted between March 2020 and June 2024, this study is a retrospective review of the registry of the State Public Health Laboratory of Jalisco, Mexico, intended to evaluate the relationships between population-level cycle threshold (Ct) values from SARS-CoV-2 real-time reverse-transcription polymerase chain reaction (RT‒PCR) tests and key COVID-19 transmission and outcome metrics for evidence of the predictive utility of Ct values for public health decision-making and early warning systems and strategies.

The work is well-written, well-defined, well-analyzed, well-referenced, and provides sufficient information for replication of the study. Additionally, limitations and suggested future research directions are offered and the Conclusions follow from the results. The weakness is that the authors have not situated their study within the body of literature regarding similar research.

Here is a Google Scholar search of the topic for research published since 2020: https://scholar.google.ca/scholar?as_ylo=2020&q=Population-level+SARS-CoV-2+RT%E2%80%92PCR+cycle+threshold+values+and+their+relationships+with+COVID%E2%80%9219+transmission+and+outcome+metrics&hl=en&as_sdt=0,5. Note that there are “About 1,450 results”. The authors must read the most relevant of these and compare and contrast their objectives to demonstrate how their study adds to the literature.

Please redo the references as per the Instructions for Authors: https://www.mdpi.com/journal/viruses/instructions.

Author Response

We sincerely thank the reviewers for their time and effort in evaluating our manuscript. Their insightful suggestions and recommendations have greatly contributed to improving the quality of this work.

We have addressed the requested corrections, and these have been highlighted in the revised manuscript for easy identification. A detailed description of the corrections is attached to this message for further reference.
Please see the attachment.

Reviewer 2 Report

Comments and Suggestions for Authors

The article “Population-level SARS-CoV-2 RT‒PCR cycle threshold values 2 and their relationships with COVID‒19 transmission and out-3 come metrics: A time series analysis across pandemic years” investigates the relationship between SARS-CoV-2 RT-PCR cycle threshold (Ct) values and key COVID-19 transmission and outcome metrics across five years of the pandemic in Jalisco, Mexico.

The article is well written and articulated, and the evidence here presented is adequate, as it correlates Ct values with significant epidemiological metrics over the pandemic years 2020-2024 in Jalisco (Mexico), supported by robust statistical analyses.

The article highlights the importance of the predictive SARS-CoV-2 Ct values in the analysed population, for monitoring COVID-19 dynamics, particularly during the early years of the pandemic 2020-2022. It underscores the importance of integrating these metrics into public health strategies and recognizing the factors that have reduced their usefulness in later stages. Overall, it provides a comprehensive framework for understanding the implications of Ct values in the context of evolving public health challenges and surveillance of infectious diseases.

However, there are limitations (partially discussed) due to the different kind of molecular tests, protocols and instruments used in the different laboratories of the selected regions, that can provide not standardized results, as well as the implications of the transition to rapid antigen tests during the last years of the pandemic. There are also missing data, for example data about vaccinated people that overall can influence the results of the study and provide interesting predictive information.

However, the article is worthy of publication on this journal.

Minor revision:

Figure 4a: The figure has not good quality, please replace it.

Author Response

(The authors gave the same response as above.)

Reviewer 3 Report

Comments and Suggestions for Authors

 De Arcos-Jiménez et al evaluated the correlation of RT-PCR Ct values as indicative of viral loads with SARS-CoV-2 infection rate, hospitalization and mortalities in a specific geographical region in Mexico. They showed that median weekly Ct values are associated  with the pandemic outcome in earlier pandemic and a weakening association in the later of the pandemic stage.

During analysis, the authors did not take into account of a major factor. Omicron has higher infectivity but less virulent than other strains. So, the Ct values  should be less in 2021-2022 and onwards. It can not be comparable with the original, alpha and delta strain as the whole pandemic. Strain specific Ct values analysis would have been useful.

Here are the following concerns

1.       In the figure 2, the positive tests results in each region, of how many patients tested would have been more informative.

2.       In table 2, the period in omicron surges, with highest infection in the world, it seems the number of positive tests markedly decreased-authors should provide an explanation in the text.

3.       In table 3, in last 3 rows, especially for COVID-19 related death, the total number of tests do not correlate with the man and woman.

4.       Figure 3 legend should be more descriptive  and self explanatory.

5.       Line 302, the decrease of death is also attributed to the mild form of the omicron than alpha or delta. It is widely recognized that omicron is less virulent than other strains although infectivities are higher. Authors should include this.

6.        

Author Response

(The authors gave the same response as above.)

Round 2

Reviewer 1 Report

Comments and Suggestions for Authors

Thank you to the authors for the changes they have made. All have improved the manuscript. Some remain.

Line by line suggested edits

83 Change “This relationship has been explored in multiple studies” to “A 2023 systematic review demonstrates the exploration of this relationship in multiple studies”.

106 “including recent reports”—Please cite these recent reports directly after this phrase rather than only regarding “most studies” in line 107 regarding citation [23].

107-109 Is this limited knowledge a product of the analysis offered in [23], or is it a finding of one of the papers reviewed for [23]? If it is a finding of one of the papers, please cite the publication directly rather than only regarding [23].

1582-1583 Please complete this reference according to the journal style rather than noting that the publication is “PubMed Available”.

1925 Please complete this reference according to the journal style rather than noting that the publication is “PubMed Available”.

1939-1940 This link does not go to the document. Please correct the link.

1942 This link does not go to the document. Please correct the link.

1945 This link does not go to the document. Please correct the link.

1954 This link does not go to the document. Please correct the link.

1959-1960 Please complete this reference according to the journal style rather than noting that the publication is “PubMed Available”.

2059 Please complete this reference according to the journal style rather than noting that the publication is “PubMed Available”.

2067 Please complete this reference according to the journal style rather than noting that the publication is “PubMed Available”.

2090 Please complete this reference according to the journal style rather than noting that the publication is “PubMed Available”.

2094 Please complete this reference according to the journal style rather than noting that the publication is “PubMed Available”.

Author Response

We thank the reviewers once again for the time and effort dedicated to our manuscript; their suggestions and recommendations have significantly enhanced the quality of this work. We have incorporated the requested corrections as part of this second round of revisions.

Attached, we provide a detailed description of the corrections made. Please see the attachment

Reviewer 3 Report

Comments and Suggestions for Authors

The text should be extensively edited and corrected.

Author Response

(The authors gave the same response as above.)
